# Phospholipase D activity couples plasma membrane endocytosis with retromer dependent recycling

**Rajan Thakur[1,2], Aniruddha Panda[1,3], Elise Coessens[4], Nikita Raj[1], Shweta Yadav[1], Sruthi Balakrishnan[1], Qifeng Zhang[4], Plamen Georgiev[4], Bishal Basak[1], Renu Pasricha[1], Michael JO Wakelam[4], Nicholas T Ktistakis[4], Padinjat Raghu[1]\***

[1]National Centre for Biological Sciences, Bangalore, India; [2]Shanmugha Arts, Science, Technology & Research Academy, Thanjavur, India; [3]Manipal University, Karnataka, India; [4]Inositide Laboratory, Babraham Institute, Cambridge, United Kingdom

**Abstract** During illumination, the light-sensitive plasma membrane (rhabdomere) of *Drosophila* photoreceptors undergoes turnover with consequent changes in size and composition. However, the mechanism by which illumination is coupled to rhabdomere turnover remains unclear. We find that photoreceptors contain a light-dependent phospholipase D (PLD) activity. During illumination, loss of PLD resulted in an enhanced reduction in rhabdomere size, accumulation of Rab7 positive, rhodopsin1-containing vesicles (RLVs) in the cell body and reduced rhodopsin protein. These phenotypes were associated with reduced levels of phosphatidic acid, the product of PLD activity and were rescued by reconstitution with catalytically active PLD. In wild-type photoreceptors, during illumination, enhanced PLD activity was sufficient to clear RLVs from the cell body by a process dependent on Arf1-GTP levels and retromer complex function. Thus, during illumination, PLD activity couples endocytosis of RLVs with their recycling to the plasma membrane thus maintaining plasma membrane size and composition.

*For correspondence: praghu@ncbs.res.in

**Competing interests:** The authors declare that no competing interests exist.

## Introduction

The ability to detect photons is a fundamental property of animal photoreceptors. In order to achieve this, ocular photoreceptors of animals generate an expanded region of plasma membrane that is packed with the receptor for light, rhodopsin. This strategy is used regardless of the architecture of the photoreceptor. For example, in ciliary photoreceptors (e.g vertebrate rods), light passes along the outer segment that is stacked with membranous discs, whereas in insect photoreceptors, the plasma membrane is expanded to form actin-based microvilli; both these structures are packed with rhodopsin, and incident light is absorbed as it passes along them (*Arendt, 2003*). The light-sensitive membranes of photoreceptors undergo stimulus-dependent turnover (*LaVail, 1976*; *White and Lord, 1975*); such turnover will alter both membrane area and composition, thus regulating sensitivity to light [reviewed in (*Blest, 1988*)]. The importance of this process is underscored by the human disease Best's macular dystrophy, in which rod outer segment length and electroretinograms are altered during changes in ambient illumination, ultimately leading to macular degeneration (*Abràmoff et al., 2013*). Despite the importance of this process, the cellular and molecular mechanisms that regulate photosensitivemembrane turnover remains poorly understood.

In *Drosophila* photoreceptors, the apical domain is expanded to form ca. 40,000 projections of light-sensitive plasma membrane (microvilli) that form the rhabdomere. Photons that are absorbed trigger G-protein-coupled phospholipase C (PLC) activity that culminates in the activation of the

**eLife digest** Certain cells in the eye contain a receptor protein known as rhodopsin that enables them to detect light. Rhodopsin is found in distinct patches on the membrane surrounding each of these "photoreceptor" cells and the number of rhodopsin molecules present controls how sensitive the cell is to light. In humans, vitamin A deficiency or genetic defects can decrease the number of rhodopsin molecules on the membrane, leading to difficulty in seeing in dim light.

Fruit fly eyes also contain rhodopsin. Exposure to normal levels of light triggers parts of the membranes of fly photoreceptor cells to detach and move into the interior of the cell. These internalized pieces of membrane have two possible fates: they can either be destroyed or recycled back to the cell surface. This membrane turnover adjusts the size of the membrane surrounding the cell and the number of rhodopsin molecules in it to regulate the cell's sensitivity to light. It is crucial that turnover is tightly regulated in order to maintain the integrity of the cell membrane. However, it is not clear how the process is regulated during light exposure.

Thakur et al. set out to address this question in fruit flies. The experiments show that an enzyme called phospholipase D is activated when photoreceptors are exposed to light. Active phospholipase D – which generates a molecule called phosphatidic acid – coordinates the internalization of pieces of membrane with the recycling of rhodopsin back to the cell surface. Thakur et al. generated fly mutants that lacked phospholipase D and in these animals the internalized rhodopsin was not transported back to the cell membrane. This caused the membrane to shrink in size and decreased the number of rhodopsin molecules in it. As a result, the photoreceptor cells became less sensitive to light.

The findings of Thakur et al. show that in response to normal levels of light, phospholipase D balances membrane internalization and recycling to maintain the size and rhodopsin composition of the membrane. Future challenges will be to work out exactly how phospholipase D is activated and how phosphatidic acid tunes membrane internalization and recycling.

plasma membrane channels TRP and TRPL; the resulting $Ca^{2+}$ influx triggers an electrical response to light (*Hardie and Raghu, 2001*). Additionally, photon absorption by rhodopsin1 (Rh1) also triggers the rhodopsin cycle [reviewed in (*Raghu et al., 2012*)]. Following photon absorption, Rh1 undergoes photoisomerization to meta-rhodopsin (M). M is phosphorylated at its C-terminus, binds β-arrestin and this complex is removed from the microvillar membrane via clathrin-dependent endocytosis to be either recycled back to the microvillar plasma membrane (*Wang et al., 2014*) or trafficked to the lysosomes for degradation (*Chinchore et al., 2009*) [reviewed in (*Xiong and Bellen, 2013*)]. Tight regulation of this process is critical for rhabdomere integrity during illumination as mutants defective in any of the several steps of the rhodopsin cycle undergo light-dependent collapse of the rhabdomere [reviewed in (*Raghu et al., 2012*)]. However, the process that couples endocytosis of rhabdomere membrane to plasma membrane recycling remains poorly understood.

Phospholipase D (PLD) is an enzyme that hydrolyzes phosphatidylcholine (PC) to generate phosphatidic acid (PA). In yeast, loss of PLD (*spo14*) results in a sporulation defect, failure to synthesize PA (*Rudge et al., 2001*) and accumulation of undocked membrane vesicles on the spindle pole body (*Nakanishi et al., 2006*). The v-SNARE Spo20p binds PA in vitro (*Nakanishi et al., 2004*) and is required to dock Spo20p to target membranes; in this setting, PA generated by PLD appears to regulate a vesicular transport process. The potential role of PA in controlling vesicular transport also arose from observations *in vitro* that Arf proteins, key regulators of vesicular transport, stimulate mammalian PLD activity (*Brown et al., 1993*; *Cockcroft et al., 1994*). Overexpression of PLD1 in a range of neuronal (*Cai et al., 2006*; *Vitale et al., 2001*) and non-neuronal cells (*Choi et al., 2002*; *Cockcroft et al., 2002*; *Huang et al., 2005*) suggests that PLD can regulate vesicular transport. A previous study showed that elevated PA levels during development of *Drosophila* photoreceptors disrupts rhabdomere biogenesis with associated endomembrane defects (*Raghu et al., 2009*) that were Arf1-dependent. However, the mechanism underlying the role of PLD in regulating membrane transport has remained unclear, and to date, no study in metazoans has demonstrated a role, if any, for endogenous PLD in regulating vesicular transport *in vivo*. In this study, we show that during

illumination in *Drosophila* photoreceptors, rhabdomere size is regulated through the turnover of apical plasma membrane via RLVs. We find that photoreceptors have a light-regulated PLD activity that is required to maintain PA levels during illumination and support apical membrane size. PLD works in coordination with retromer function and Arf1 activity to regulate apical membrane size during illumination. Thus, PLD is a key regulator of plasma membrane turnover during receptor activation and signaling in photoreceptors.

## Results

### Rhabdomere size and Rh1 levels are modulated by illumination in *Drosophila*

We quantified rhabdomere size of *Drosophila* photoreceptors during illumination by transmission electron microscopy (TEM) followed by volume fraction analysis. When wild-type flies are grown in white light for 48 hr (hrs) post-eclosion, the volume fraction ($V_f$) of the cell occupied by the rhabdomere in peripheral photoreceptors R1-R6 was reduced (*Figure 1A,B*). This reduction in $V_f$ occurred prior to the onset of any obvious vesiculation or rhabdomere degeneration; the $V_f$ of rhabdomere R7 that expresses UV-sensitive rhodopsin (that does not absorb white light) did not change (*Figure 1A,B*). This reduction in rhabdomere size was accompanied by changes in the localization of Rh1, the rhodopsin isoform expressed in R1-R6. With just 12 hr of illumination, there was an increase in the number of RLVs in the cell body (*Figure 1C,D*). A subset of these RLVs co-localize with the early and late endocytic compartment markers Rab5 and Rab7, respectively (*Figure 1E,F*). Over a period of 4 days, illumination results in a reduction in total Rh1 protein levels (*Figure 1G*) and manifests functionally as a reduction in sensitivity to light (*Figure 1H*).

### *dPLD* is required to support rhabdomere volume during illumination

We generated loss-of-function mutants in *dPLD* using homologous recombination (*Gong and Golic, 2003*) (*Figure 2—figure supplement 1A*). Multiple alleles were isolated of which *dPLD$^{3.1}$* is described in detail. To test if *dPLD$^{3.1}$* represents an animal with no residual PLD activity, we used the transphosphatidylation assay that exploits the ability of PLD to use primary alcohols as nucleophilic acceptor. Flies were starved for 12 hr, allowed to feed for 6 hr on 10% ethanol/sucrose and the formation of phosphatidylethanol (PEth) was monitored using LC-MS (*Wakelam et al., 2007*). Under these conditions, multiple species of PEth were detected in wild-type flies, no PEth could be detected in *dPLD$^{3.1}$* extracts under the equivalent conditions (*Figure 2—figure supplement 1C,D*). Thus, *dPLD$^{3.1}$* mutants have no residual PLD activity.

*dPLD$^{3.1}$* flies are homozygous viable as adults. At eclosion, photoreceptor ultrastructure in *dPLD$^{3.1}$* was indistinguishable from controls (*Figure 2A*). Following exposure to 2000 lux white light for 48 hr, as expected, $V_f$ occupied by peripheral rhabdomeres was reduced in wild-type flies (*Figure 2B*), whereas $V_f$ of R7 was unaffected. In *dPLD$^{3.1}$*, rhabdomere $V_f$ reduced following illumination (*Figure 2B*); however, the reduction was substantially greater than in wild type (*Figure 2C*).

We visualized RLVs in photoreceptors by Rh1 immunolabeling and counted them. These analyses were done at 0 days post-eclosion, prior to the onset of any obvious ultrastructural change in *dPLD$^{3.1}$*. In dark-reared flies, the number of RLVs in *dPLD$^{3.1}$* was greater than in wild-type photoreceptors (*Figure 2D,E*). Following illumination for 12 hr, the number of RLVs increases in both controls and *dPLD$^{3.1}$*; however, the increase was greater in *dPLD$^{3.1}$* (*Figure 2E*). Further, while the number of RLVs that were Rab5-positive was not significantly different between controls and *dPLD$^{3.1}$*, the number of Rab7-positive RLVs were significantly greater in *dPLD$^{3.1}$* compared to controls (*Figure 2F*). Thus, during illumination there is enhanced accumulation of RLVs in a Rab7 positive compartment in *dPLD$^{3.1}$*.

We measured Rh1 protein levels using Western blotting in flies exposed to bright illumination for four days post-eclosion. As expected, levels of Rh1 decreased when wild-type flies were reared in bright light compared to dark-reared controls (*Figure 2G*). In dark reared flies, Rh1 levels are equivalent in controls and *dPLD$^{3.1}$* (*Figure 2G*); following illumination Rh1 levels decrease in both genotypes but the reduction seen in *dPLD$^{3.1}$* is much greater than in wild-type flies of matched eye color (*Figure 2G,H*). Consistent with this, we found that *dPLD$^{3.1}$* photoreceptors were less sensitive to light compared to controls of matched eye color on eclosion (*Figure 2J*). These findings

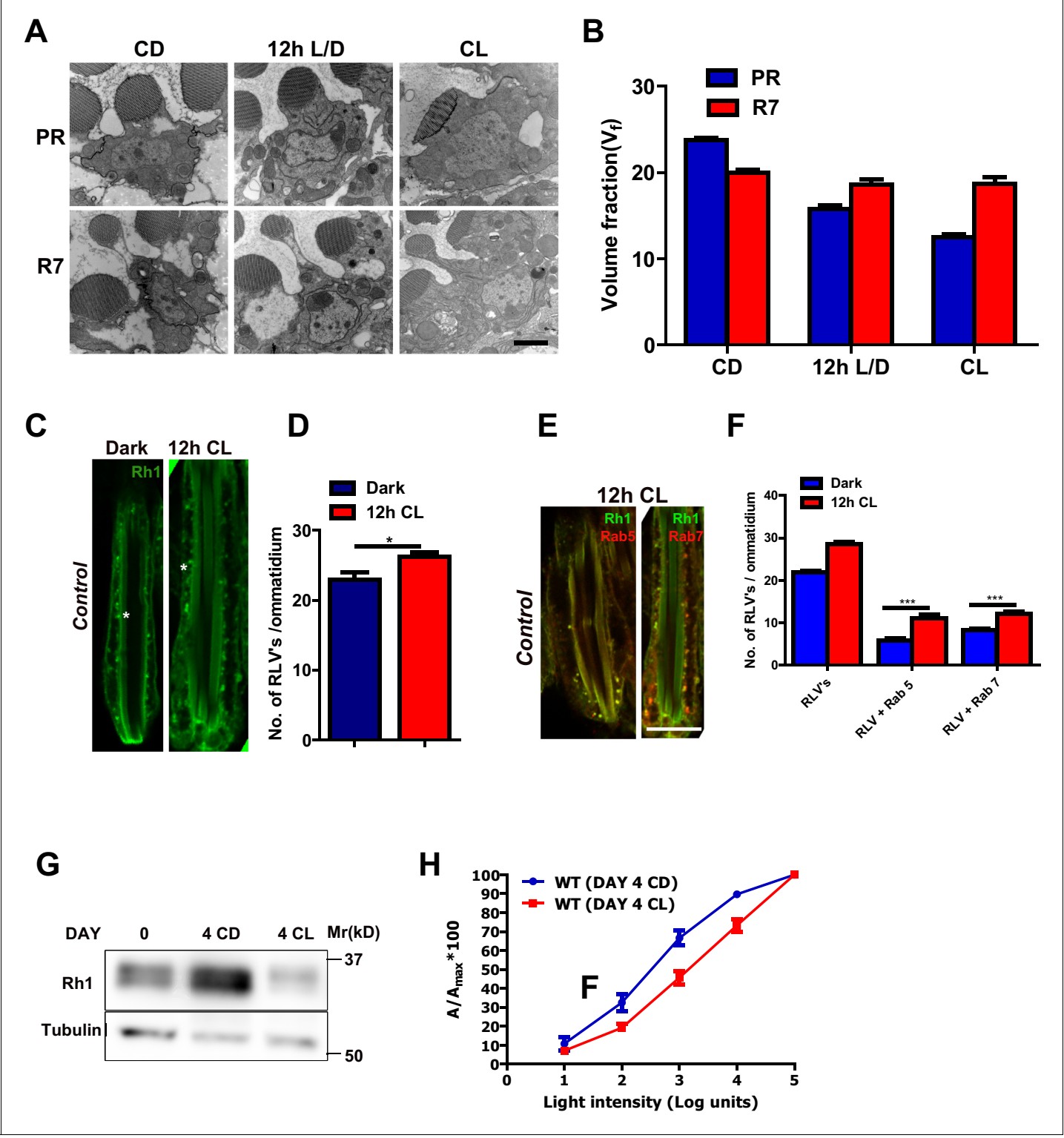

**Figure 1.** Rhabdomere size regulation during illumination in *Drosophila* photoreceptors. (A) TEM images of single rhabdomere from wild-type photoreceptors (PRs) of 2-day-old flies post-eclosion reared in constant dark (CD), 12 hr light, 12 hr dark (12 h L/D) and constant light (CL). Scale bar: 1 µm. (B) Quantification of rhabdomere volume in PRs reared in various conditions. The peripheral PRs represent R1 to R6 rhabdomeres. The X-axis represents the rearing condition and the Y-axis represents the volume fraction ($V_f$) of rhabdomere expressed as a % with respect to total cell volume. n = 90 rhabdomeres taken from three separate flies. (C) Longitudinal section (LS) of retinae from control stained with rhodopsin 1 (Rh1) antibody. Flies were dissected after 0–6 hr (day 0) and 12 hr of bright light illumination (12 h CL) post-eclosion. Scale bar: 5 µm. (D) Quantification of RLVs from LS of *Figure 1 continued on next page*

Figure 1 continued

retinae from control. The X-axis represents the time point and rearing condition. Y-axis shows the number of RLV's per ommatidium. n = 10 ommatidia taken from three separate preps. (E) LS of retinae from control stained with Rh1 and Rab5; Rh1 and GFP (for *Rh1>GFP::Rab7*). Rearing condition is same as mentioned in (panel C). Scale bar: 5 µm. (F) Quantification of RLVs from LS of retinae from control. The X-axis represents the population of vesicles positive for mentioned protein. Y-axis shows the number of RLVs per ommatidium. n = 10 ommatidia taken from three separate preps. (G) Western blot from head extracts of control flies reared in various conditions as indicated on the top of the blot. The blot was probed with antibody to rhodopsin. Tubulin levels were used as a loading control. (H) Intensity response function of the light response from 4-day-constant-light (DAY 4 CL) and 4-day-constant-dark (DAY 4 CD) old control flies. The X-axis represents increasing light intensity in log units and Y-axis the peak response amplitude at each intensity normalized to the response at the maximum intensity. n=separate flies. Data are presented as mean ± SEM.

demonstrate that during illumination, the turnover of Rh1, an apical membrane protein of photoreceptors is altered in $dPLD^{3.1}$. However, such changes were not seen in the levels or localization of TRP and NORPA, two other apical membrane proteins, during illumination (**Figure 2—figure supplement 2A,B**). $dPLD^{3.1}$ photoreceptors did not exhibit a primary defect in the electrical response to light in electroretinograms (**Figure 2I, Figure 2—figure supplement 3 A-D**).

## Retinal degeneration in $dPLD^{3.1}$ is dependent on altered PA levels

We grew flies in constant illumination following eclosion. Under these conditions, control photoreceptors maintain normal structure; however, $dPLD^{3.1}$ undergoes light-dependent retinal degeneration. The degeneration starts by day 5 post-eclosion and all six peripheral photoreceptors degenerate by day 14 (**Figure 3A,B**). This degeneration is strictly dependent on illumination as $dPLD^{3.1}$ not exposed to light retains normal ultrastructure up to day 14 (**Figure 3A,B**). Photoreceptor degeneration is underpinned by a collapse of the apical microvillar membrane as well as the accumulation of membranous whorls within the cell body (**Figure 3C**). Retinal degeneration was also seen in a trans-heterozygote combination of two independently isolated alleles $dPLD^{3.1}$ and $dPLD^{3.3}$ (**Figure 3—figure supplement 1A, B**). No degeneration was seen in either $dPLD^{3.1}/+$ or $dPLD^{3.3}/+$ (**Figure 3—figure supplement 1A**) excluding a dominant negative or neomorphic effect of these alleles. The light-dependent degeneration was also seen when the $dPLD^{3.1}$ allele was placed over a deficiency chromosome for the $dPLD$ gene region; in $dPLD^{3.1}/Df(2R)ED1612$, retinal degeneration was comparable and was no worse than in $dPLD^{3.1}$ homozygotes (**Figure 3—figure supplement 1C**), suggesting that $dPLD^{3.1}$ is a null allele. Light-dependent degeneration in $dPLD^{3.1}$ could be rescued by a wild-type transgene [$dPLD^{3.1};Hs>dPLD$] but not by a lipase dead transgene [$dPLD^{3.1};Hs>dPLD^{K/R}$] (**Figure 3D,E,F**). These results demonstrate that dPLD enzyme activity is required to support normal photoreceptor ultrastructure during illumination.

In order to understand the biochemical basis of retinal degeneration of $dPLD^{3.1}$, we measured levels of PC and PA from retinal extracts using direct infusion mass spectrometry (**Schwudke et al., 2011**). We found that levels of PC were not significantly different between controls and $dPLD^{3.1}$ (**Figure 4A**). By contrast, there was a significant decrease in total PA levels in $dPLD^{3.1}$ (**Figure 4B**). At the level of molecular species, this reduction was associated with changes in the levels of PA species with longer acyl chain lengths (**Figure 4C**). Rescue of retinal degeneration in $dPLD^{3.1}$ by reconstitution with $Hs>dPLD$ was associated with restoration in PA levels back to that of controls (**Figure 4D**). Reconstitution with $Hs>dPLD^{K/R}$ that failed to rescue degeneration also did not restore PA levels in $dPLD^{3.1}$ (**Figure 4D**). These results show that retinal degeneration in $dPLD^{3.1}$ is correlated with reduced PA levels.

We hypothesized that if the retinal degeneration in $dPLD^{3.1}$ is due to reduced PA levels, elevating PA levels in $dPLD^{3.1}$ retinae by methods independent of dPLD activity should rescue this phenotype. It is reported that in $laza^{22}$ photoreceptors, lacking Type II PA phosphatase activity, PA levels rise during exposure to light (**Garcia-Murillas et al., 2006**). We generated double mutants $dPLD^{3.1};laza^{22}$ and studied retinal degeneration in these flies. We found that $dPLD^{3.1};laza^{22}$ photoreceptors did not undergo light dependent-degeneration (**Figure 4E**). To test if this was due to restoration of PA levels, we measured PA levels in all these genotypes. As previously reported, we found that PA levels were elevated in $laza^{22}$; importantly, the reduced levels of PA seen in $dPLD^{3.1}$ was restored in $dPLD^{3.1};laza^{22}$ (**Figure 4F**). We also overexpressed $rdgA$, encoding the major diacylglycerol kinase (DGK) activity in photoreceptors. Overexpression of $rdgA$ has previously been shown to

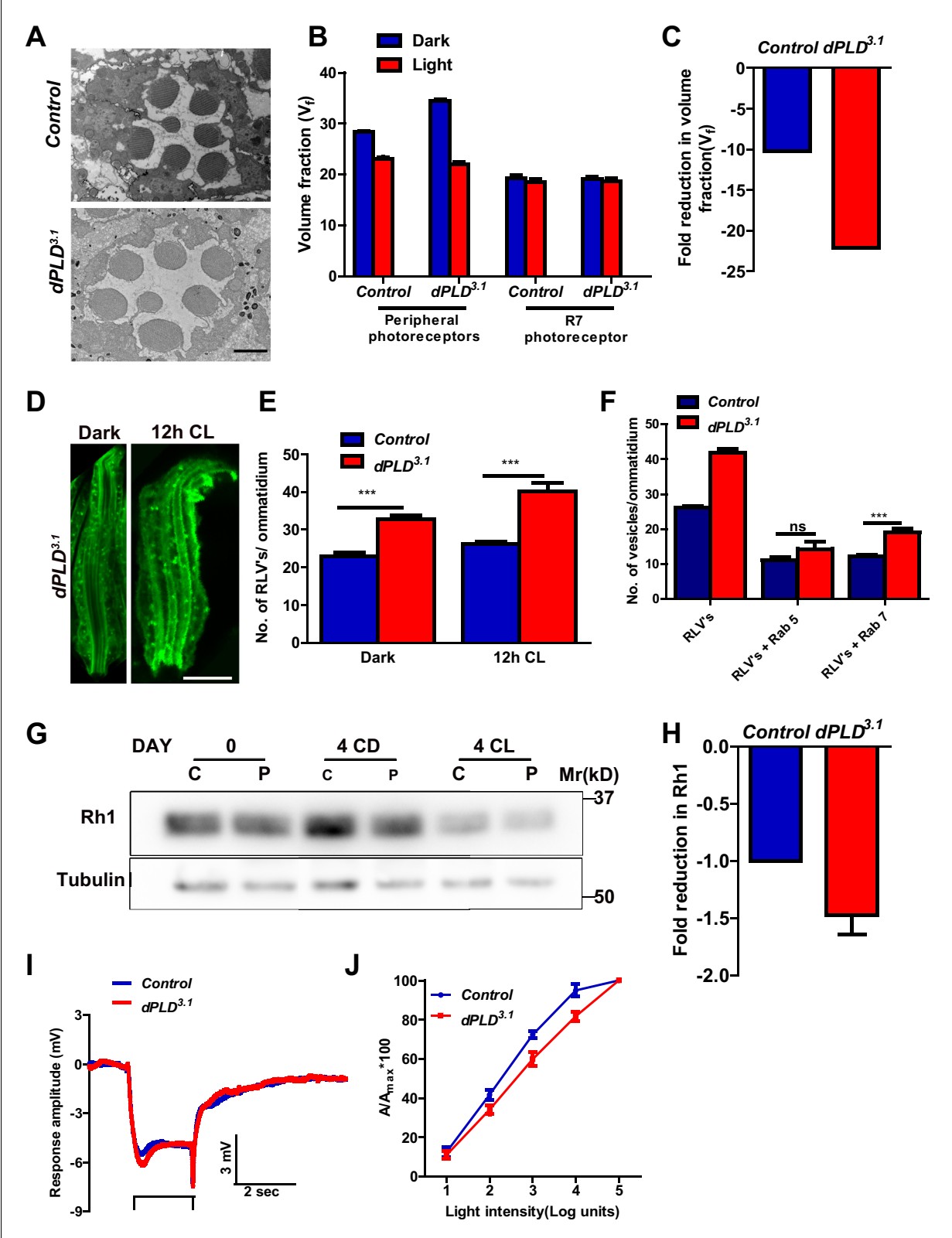

**Figure 2.** *dPLD* is required to support rhabdomere volume during illumination. (A) TEM images showing single ommatidium from control and *dPLD³·¹*. PRs of 0- to 12-hr-old flies post eclosion. Scale bar: 1 μm (B) Quantification of the rhabdomere volume of control and *dPLD³·¹*. PRs reared in constant dark and constant light for 2 days post-eclosion. n = 90 rhabdomeres taken from three separate flies. (C) Quantification of fold reduction in rhabdomere volume of control and *dPLD³·¹* in light compared to dark. Genotypes are indicated on the X-axis and the Y-axis represents the percentage

*Figure 2 continued on next page*

Figure 2 continued

volume fraction ($V_f$) of the rhabdomere with respect to cell. (**D**) LS of retinae stained with rhodopsin one from *dPLD*[3.1]. Rearing conditions are indicated. Scale bar: 5 µm. (**E**) Quantification of RLVs from LS of retinae from control and *dPLD*[3.1]. n = 10 ommatidia taken from three separate preps. (**F**) Quantification of RLVs from LS of retinae from control and *dPLD*[3.1] reared in 12 hr CL. The X-axis represents the population of vesicles positive for mentioned protein. Y-axis shows the number of RLVs per ommatidium. n = 10 ommatidia taken from three separate preps. (**G**) Western blot from head extracts of control (C) and *dPLD*[3.1] (P) of matched eye color. Rearing conditions as indicated on the top of the blot. The blot was probed with antibody to rhodopsin. Tubulin levels were used as a loading control. (**H**) Quantification of fold reduction of rhodopsin seen in *dPLD*[3.1] normalized to controls. The X-axis shows the genotype. Y-axis represents the fold reduction in rhodopsin. n = 3. (**I**) Representative ERG responses of 0- to 12-hr-old flies to a single 2 s flash of green light. Genotypes are indicated. X-axis represents the time in seconds (s) and the Y-axis represents the amplitude of response in mV. The duration of light pulse is indicated. (**J**) Intensity response function of the light response of 0- to 12-hr-old flies. Responses from control and *dPLD*[3.1] flies with matched eye color are shown. The X-axis represents increasing light intensity in log units and Y-axis the peak response amplitude at each intensity normalized to the response at the maximum intensity. n= five separate flies. Data are presented as mean ± SEM.

The following figure supplements are available for figure 2:

**Figure supplement 1.** Characterization of *dPLD*[3.1] loss-of-function allele.

**Figure supplement 2.** Normal levels and localization of other apical domain proteins in *dPLD*[3.1].

**Figure supplement 3.** Electrophysiological characterization of *dPLD*[3.1].

elevate PA levels without affecting retinal ultrastructure (*Raghu et al., 2009*). When *rdgA* is overexpressed in *dPLD*[3.1] (*dPLD*[3.1];*Rh1>rdgA*), retinal degeneration was completely rescued and the reduced PA levels seen in *dPLD*[3.1] were reverted back to wild type levels (*Figure 4—figure supplement 1A,B*). Collectively, these observations suggest that reduced PA levels underlie the retinal degeneration phenotype of *dPLD*[3.1].

## Illumination-dependent dPLD activity regulates PA levels and Rh1 turnover

The finding that *dPLD*[3.1] undergoes light-dependent retinal degeneration suggests that dPLD might be activated during illumination. When *Drosophila* photoreceptors are illuminated, a key source of PA is the sequential activity of PLCβ and DGK (*Inoue et al., 1989*; *Yoshioka et al., 1983*) and PA is also metabolized by the PA phosphatase *laza* (*Garcia-Murillas et al., 2006*). In order to uncover a potential *dPLD* generated pool of PA, we exploited *dGq*[1] mutants in which the failure to activate PLCβ results in a suppression of PA production via DGK (*Garcia-Murillas et al., 2006*). We compared PA levels in retinal extracts from *dGq*[1] with *dGq*[1],*dPLD*[3.1] both in the dark and following illumination with 12 hr of light. PA levels from both genotypes were comparable in dark reared flies; however, PA levels rise in *dGq*[1] mutants following illumination presumably reflecting production from a non-PLCβ-DGK source (*Figure 4G*). This rise in PA levels was suppressed in *dGq*[1], *dPLD*[3.1] flies (*Figure 4G*). Thus, illumination induces dPLD dependent PA production in *Drosophila* photoreceptors.

*dPLD* was overexpressed in adult photoreceptors (*Rh1>dPLD*). Following 12 hr of white light illumination, the number of RLVs increases in the cell body of wild type (*Figure 4H,I*). However, in *Rh1>dPLD* the number of RLVs did not increase (*Figure 4H,I*); this effect was not seen on overexpression of *Rh1>dPLD*[K/R] (*Figure 4H,I*) suggesting that the ability of *dPLD* to regulate RLV turnover is dependent on its catalytic activity. Together, these observations suggest that during illumination *dPLD* activity can support RLVs turnover.

## dPLD activity supports RLV removal from the cell body during illumination

RLV numbers in the cell body are an outcome of the balance between ongoing clathrin- dependent endocytosis of Rh1 containing rhabdomere membrane as well as mechanisms that remove these from the cell body. To understand the mechanism underlying the increased RLV number in *dPLD*[3.1], we exploited the temperature-sensitive allele of dynamin, *shi*[ts1]. At the permissive temperature of 18°C, where dynamin function is normal, we exposed flies to a 5-min pulse of bright white light to

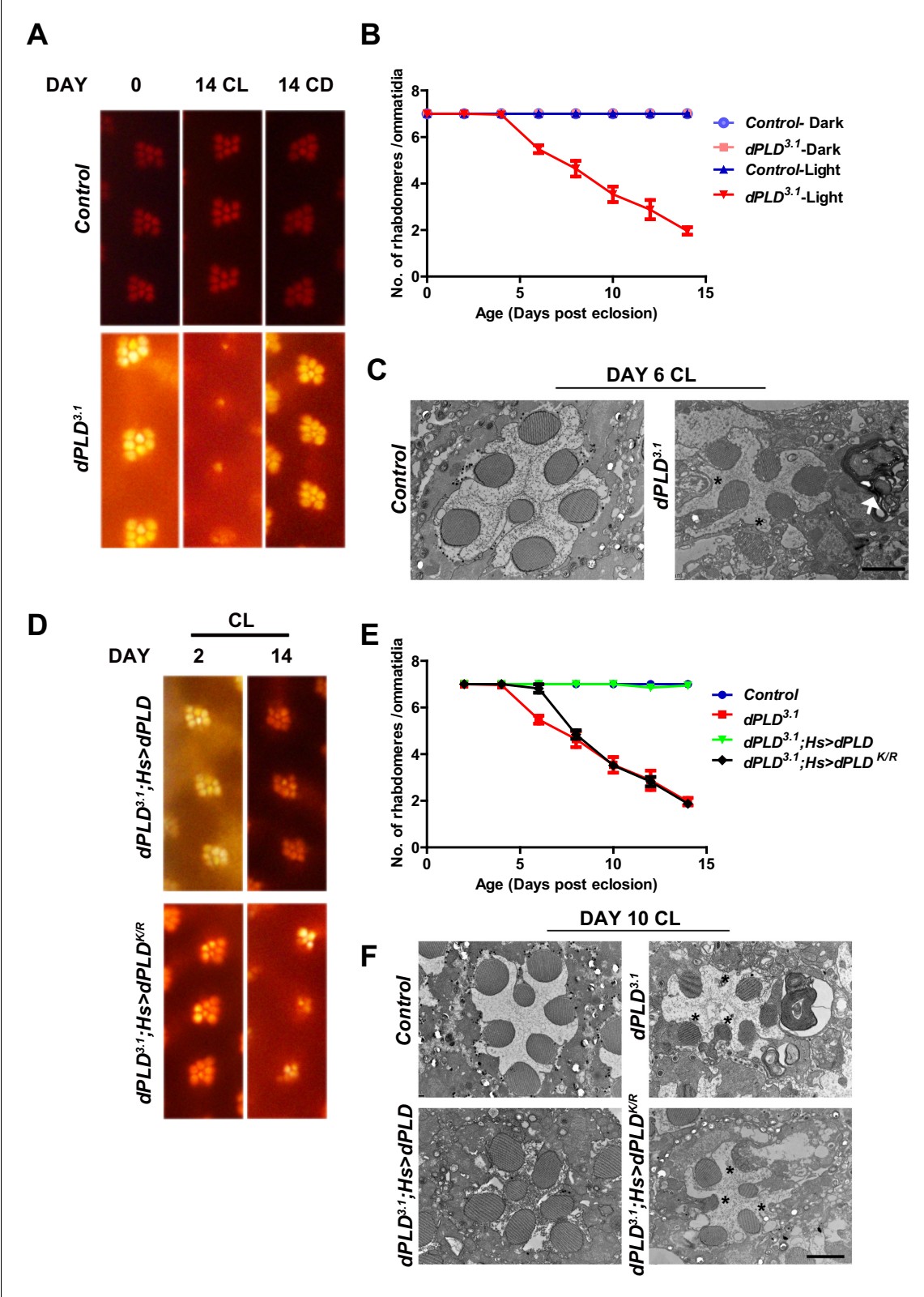

**Figure 3.** *dPLD* is essential to support rhabdomere structure during illumination. (**A**) Representative optical neutralization (ON) images showing rhabdomere structure from control and *dPLD*[3.1]. The age and rearing conditions are mentioned on the top of the panels. (**B**) Quantification of rate of PR degeneration of control and *dPLD*[3.1] reared in bright light. The X-axis represents age of the flies and the Y-axis represents the number of intact rhabdomeres visualized in each ommatidium. n = 50 ommatidia taken from at least five separate flies. (**C**) TEM images showing a single ommatidium

*Figure 3 continued on next page*

*Figure 3 continued*

from control and *dPLD$^{3.1}$* PRs reared in bright illumination for 6 days post eclosion. * indicates the collapsed rhabdomere and the arrow head indicate whorl like membranes accumulated in the cell body. Scale bar 1 μm. (D) Representative ON images showing ommatidia from *dPLD$^{3.1}$;Hs>dPLD* and *dPLD$^{3.1}$;Hs>dPLD$^{K/R}$*. The age and rearing conditions are indicated on the top of the image. (E) Quantification of rate of PR degeneration of control, *dPLD$^{3.1}$*, *dPLD$^{3.1}$;Hs>dPLD* and *dPLD$^{3.1}$;Hs>dPLD$^{K/R}$* reared in bright light. n = 50 ommatidia taken from at least five separate flies. (F) TEM images showing a single ommatidium from control, *dPLD$^{3.1}$*, *dPLD$^{3.1}$;Hs>dPLD* and *dPLD$^{3.1}$;Hs>dPLD$^{K/R}$* PRs reared in light for 10 days post-eclosion. Scale bar 1 μm. Data are presented as mean ± SEM.

The following figure supplement is available for figure 3:

**Figure supplement 1.** Genetic validation of the retinal degeneration phenotype of *dPLD$^{3.1}$*.

trigger Rh1 isomerization to M and trigger its endocytosis. Under these conditions, the number of RLVs generated in cells with and without PLD function was indistinguishable (*Figure 5A,B*). Following this, animals were rapidly shifted to 25°C, incubated for various time periods, retinae were fixed, processed and RLVs counted. Under these conditions, in *shi$^{ts1}$*, where there is no further ongoing endocytosis, RLV numbers fall rapidly, presumably reflecting the removal of previously endocytosed vesicles (*Figure 5A*). By contrast, in *shi$^{ts1}$;dPLD$^{3.1}$*, following the shift to 25°C post-illumination, there was no drop in RLV number with time implying a defect in mechanisms that remove RLVs from the cell body (*Figure 5B*).

We counted RLVs in *norpA$^{P24}$* subjected to bright illumination; as previously reported, we found RLV numbers were elevated (*Chinchore et al., 2009*). This elevation in RLV number could be suppressed by the overexpression of *dPLD* (*Figure 5C,D*). We also found that the light-dependent retinal degeneration in *norpA$^{P24}$* that is reported to depend on RLV accumulation in a Rab7 compartment (*Chinchore et al., 2009*) (*Wang et al., 2014*) could be partially suppressed by overexpressing *dPLD* (*Figure 5E,F*). Interestingly, we found that during illumination, in *Rh1>dPLD*, there was a significant reduction in the number of Rab7-positive RLVs but not in the number of Rab5-positive RLVs (*Figure 5G*). Collectively, these findings show that *dPLD* supports a process that can clear RLVs from the cell body of photoreceptors during illumination.

## dPLD regulates clearance of RLVs via retromer function

The retromer complex plays a central role in removing endocytosed transmembrane proteins from the lysosomal pathway and targets them to other cellular compartments (*Gallon and Cullen, 2015*). We tested the effect of manipulating core members of the retromer complex in photoreceptors. RNAi downregulation of *vps35* results in an increase in RLV numbers both in the dark and following 12 hr illumination (*Figure 6A,B*). We tested the effect of overexpressing *vps35* in photoreceptors during illumination; in an otherwise wild-type fly, this did not result in changes in RLV number (*Figure 6D*) or caused retinal degeneration (*Figure 6C*). However, in *dPLD$^{3.1}$* photoreceptors, overexpression of *vps35* results in two key outcomes: (i) the increased numbers of RLVs seen in *dPLD$^{3.1}$* are reduced back to wild type levels (*Figure 6D*) and (ii) the retinal degeneration of *dPLD$^{3.1}$* is suppressed (*Figure 6C*).

During illumination, overexpression of *dPLD* results in a reduction of RLV number in a lipase-dependent manner (*Figure 4H,I*). We tested the requirement of intact retromer function for the ability of *dPLD* to clear RLVs. We found that in cells where *vps35* was downregulated, overexpression of *dPLD* could not reduce RLV numbers (*Figure 6E,F*). These findings suggest that intact retromer function is required for *dPLD* to support the clearance of RLVs during illumination.

## Overexpression of *garz* rescues retinal degeneration in *dPLD$^{3.1}$*

We overexpressed *garz*, the *Drosophila* ortholog of the guanine nucleotide exchange factor (GBF1) of Arf1 (*Cox et al., 2004*). In adult photoreceptors, *garz* overexpression does not impact rhabdomere structure during illumination (*Figure 7B,C*), although RLV numbers were reduced (*Figure 7A*). When *garz* is overexpressed in *dPLD$^{3.1}$*, it completely rescues retinal degeneration (*Figure 7B,C*). These findings strongly suggest that retinal degeneration in *dPLD$^{3.1}$* may be due to reduced ARF1 activity. If this model is true then reducing *garz* activity in wild-type flies should phenocopy *dPLD$^{3.1}$*. To test this, we down-regulated *garz* in photoreceptors; this resulted in light-dependent retinal

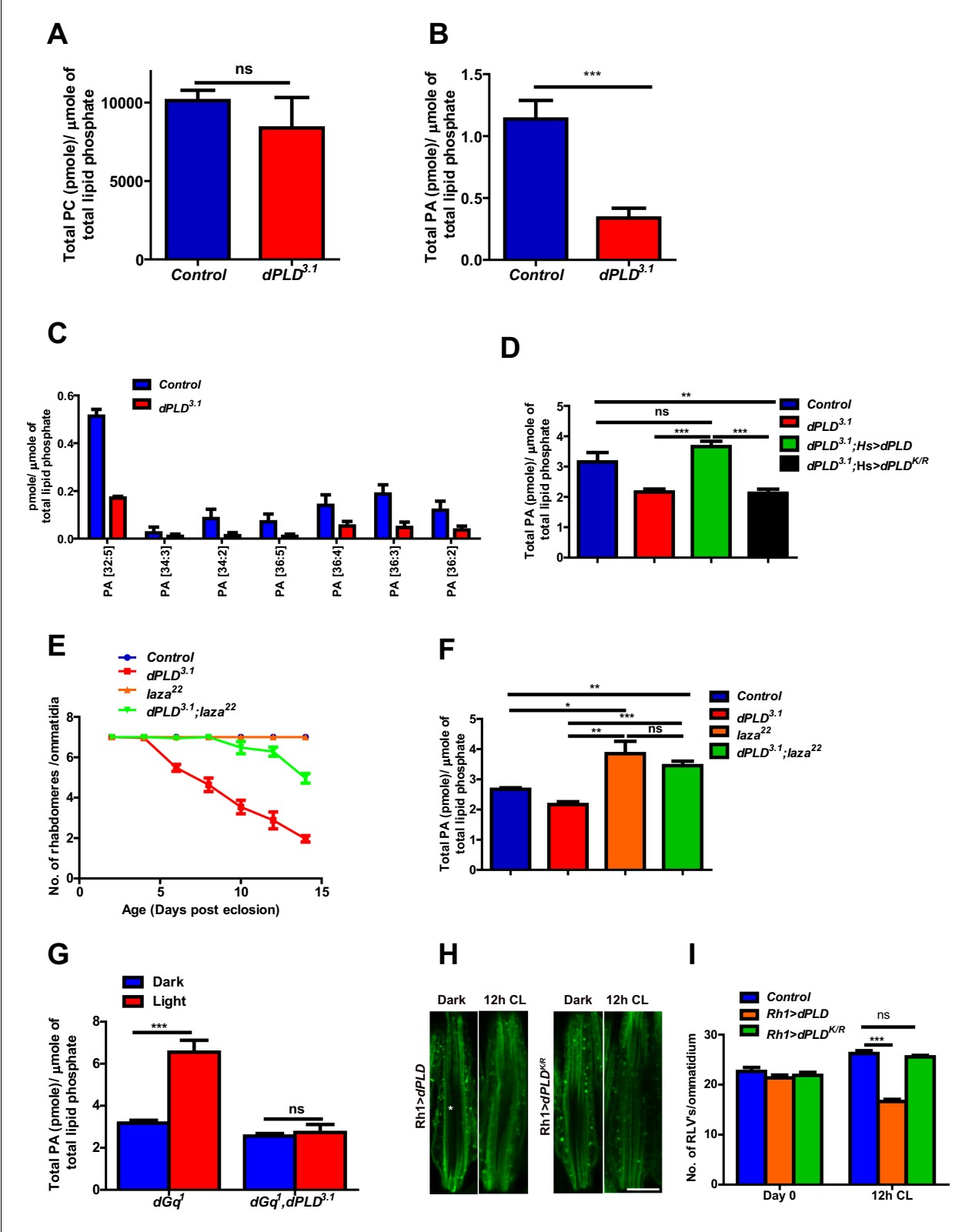

**Figure 4.** Phosphatidic acid levels and retinal degeneration in *dPLD*[3.1.] (**A**) Total PC level in retinae of control and *dPLD*[3.1]. The X-axis represents the genotypes and the Y-axis shows the level of PC as pmole/μmole of total lipid phosphate present in the sample. n = 3. (**B**) Total PA level in retinae of control and *dPLD*[3.1]. The X-axis represents the genotypes and the Y-axis shows the level of PA as pmole/μmole of total lipid phosphate present in the sample. n = 3. (**C**) Molecular species of PA in retinae of control and *dPLD*[3.1]. X-axis shows the acyl chain composition of each species predicted from its

*Figure 4 continued on next page*

Figure 4 continued

monoisotopic peaks and corresponding elemental composition constraints. Y-axis shows the abundance of each species as pmole/µmole of total lipid phosphate present in the sample. n = 3. (D) PA levels in heads extracts of control, *dPLD*[3.1], *dPLD*[3.1];*Hs>dPLD* and *dPLD*[3.1];*Hs>dPLD*[K/R] . n = 3. (E) Quantification of retinal degeneration seen in control, *laza*[22], *dPLD*[3.1] and *dPLD*[3.1];*laza*[22]. n= 50 ommatidia taken from at least five separate flies. (F) PA levels in heads extracts of control, *laza*[22], *dPLD*[3.1] and *dPLD*[3.1];*laza*[22] n = 3. (G) PA levels from retinal extracts of *Gq*[1] and *Gq*[1],*dPLD*[3.1]. Flies were reared in complete darkness and post ecclosion one set of flies were shifted to bright illumination for 12 hr while the others kept in darkness for 12 hr. n = 3. (H) LS of retinae stained with Rh1 from *Rh1>dPLD* and *Rh1>dPLD*[K/R]. Rearing conditions are indicated at the top of panels. Scale bar:5 µm. (I) Quantification of RLVs from LS of retinae from control, *Rh1>dPLD* and *Rh1>dPLD*[K/R]. n = 10 ommatidia taken from three separate preps. Data are presented as mean ± SEM.

The following figure supplement is available for figure 4:

**Figure supplement 1.** Rescue of *dPLD*[3.1] phenotypes by overexpression of DGK.

degeneration, the kinetics of which were comparable to that of *dPLD*[3.1] (*Figure 7D,E*). Finally, we found that overexpression of *garz* reduced RLV number in *dPLD*[3.1] back toward wild-type controls (*Figure 7F*).

## *dPLD* and *garz* are required for RLV clearance during illumination

Since both *garz* and *dPLD* play a role in RLV clearance during illumination (*Figure 7* and *Figure 4*), we tested the requirement of each molecule on the other for this function. We found that the ability of Rh1>dPLD to clear RLVs required intact *garz* function. When *garz* is also depleted (*Rh1>garz*[RNAi]) in *Rh1>dPLD* cells, the reduction in RLV number seen in *Rh1>dPLD* alone was attenuated (*Figure 8A*). This finding suggests that a *garz*-dependent step is required to support RLV clearance by dPLD during illumination.

We also explored the route by which *garz* activity clears RLVs. When *Rh1>garz* is performed in photoreceptors where retromer function is depleted (*Rh1>vps35*[RNAi]), the reduction in RLV number seen in *Rh1>garz* alone is substantially blocked (*Figure 8B*). Thus, the ability of *garz* to support RLV clearance from the cell body requires intact retromer function.

Since our observations indicate a role of dPLD and its product PA in the context of Arf1-GTP activity, we tested the requirement for dPLD in regulating the biological activity of Arf1. In photoreceptors, overexpression of constitutively active Arf1, Arf1[CA] (*Rh1>Arf1*[CA]), results in ultrastructure defects in the rhabdomere (*Figure 8C*). We expressed *Rh1>Arf1*[CA] in *dPLD*[3.1] and studied its effect on ultrastructure. In the absence of dPLD function, the effect of *Rh1>Arf1*[CA] on ultrastructure was substantially reduced (*Figure 8C* iii versus iv). This finding suggests that PA produced by dPLD is required to mediate the effects of Arf1 *in vivo*.

## Discussion

Although the importance of plasma membrane turnover in determining cellular responses to external stimuli is well appreciated, the mechanisms that regulate this process remain unclear. In photoreceptors, change in size of photosensitive membranes during illumination represents a special example of the broad principle of plasma membrane turnover following receptor-ligand interaction. In *dPLD*[3.1] photoreceptors, the process of light-induced membrane turnover is exaggerated; these photoreceptors show larger reductions in rhabdomere volume and greater reductions in Rh1 levels than is seen in wild-type flies. The physiological consequence of this is that *dPLD*[3.1] photoreceptors are less sensitive to light than controls when reared in light (*Figure 2J*).

During illumination, RLVs are generated and mature through Rab5 and Rab7 endocytic compartments. We found that (i) photoreceptors contain a light-stimulated PLD activity, (ii) Loss of dPLD activity results in enhanced numbers of Rab7-positive RLVs in the cell body during illumination, (iii) overexpression of catalytically active dPLD was able to clear light-induced Rab7-positive RLVs in wild-type cells and (iv) dPLD overexpression was able to reduce the enhanced RLV number and partially suppress retinal degeneration in *norpA*[P24], a mutant that shows enhanced Rab7 positive RLVs during illumination. Thus ,dPLD represents an enzyme activity that couples the generation of RLVs by light-induced endocytosis to their removal from the cell body. It has previously been reported

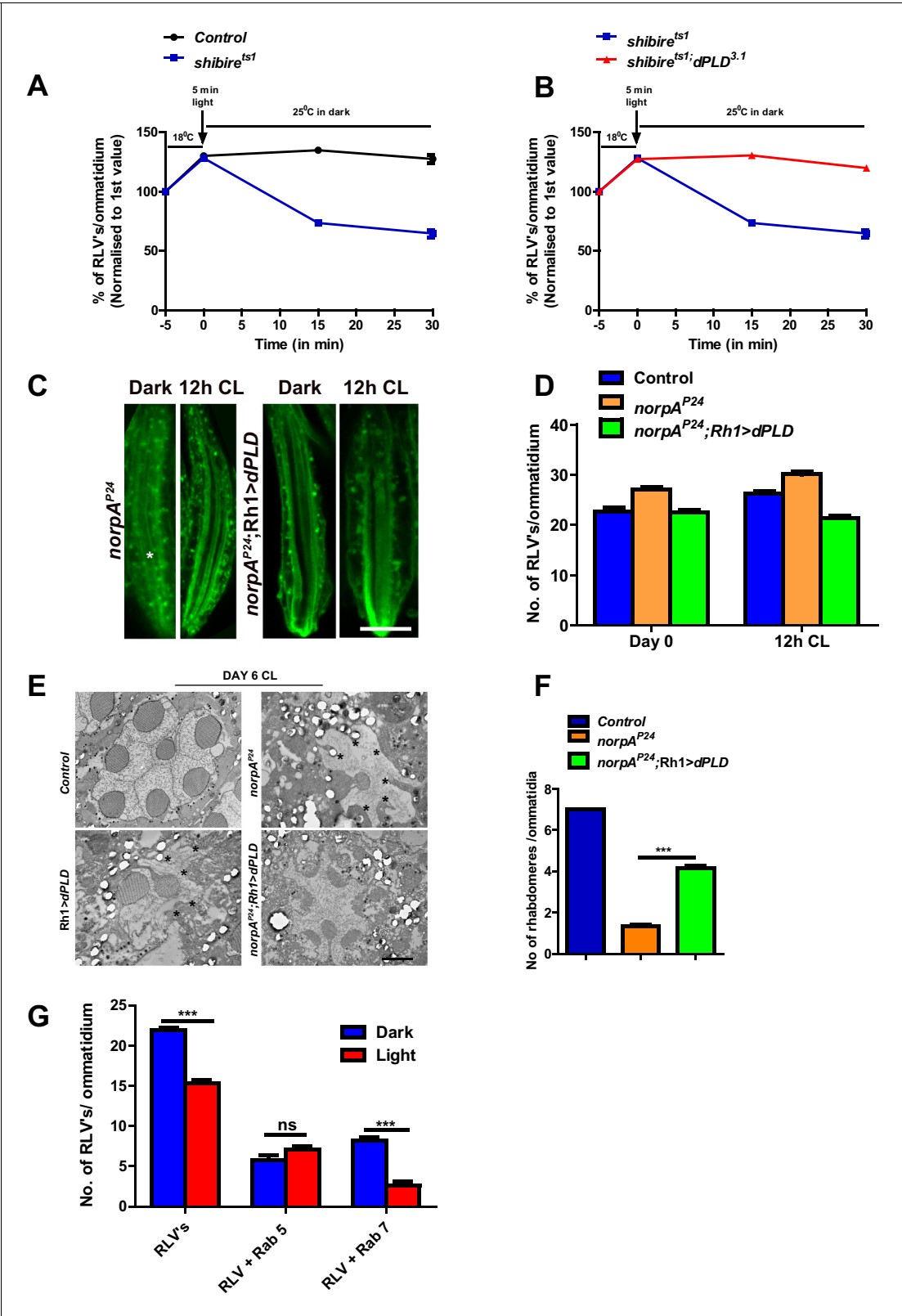

**Figure 5.** dPLD activity supports the removal of RLVs from the cell body during illumination. (A) Quantification of RLVs from LS of retinae from control and *shi*[ts1]. n = 10 ommatidia taken from three separate preps. (B) Quantification of RLVs from LS of retinae from *shi*[ts1] and *shi*[ts1];*dPLD*[3.1]. n = 10 ommatidia taken from three separate preps. (C) LS of retinae stained with Rh1 from *norpA*[P24] and *norpA*[P24];*Rh1>dPLD*. Rearing condition is indicated at the top of each panel. Scale bar: 5 μm. (D) Quantification of RLVs from LS of retinae from control, *norpA*[P24] and *norpA*[P24];*Rh1>dPLD*. n = 10 ommatidia

*Figure 5 continued on next page*

*Figure 5 continued*

taken from three separate preps. (**E**) TEM images showing single ommatidium from control, *norpA^P24^*, *Rh1>dPLD* and *norpA^P24^;Rh1>dPLD* PRs of flies.
* indicates the degenerated rhabdomere. Rearing condition is indicated on the top of the image. Scale bar: 1 µm. (**F**) Quantification of retinal degeneration in control, *norpA^P24^* and *norpA^P24^;Rh1>dPLD* done using TEM images. The Y-axis represents the number of rhabdomeres visualized in each ommatidium. n = 50 ommatidia taken from at least two separate flies. (**G**) Quantification of RLVs from LS of retinae from *Rh1>dPLD* in dark vs light (12 h CL). The X-axis represents the population of vesicles positive for mentioned protein. Y-axis shows the number of RLV's per ommatidium. n = 10 ommatidia taken from three separate preps. Data are presented as mean ± SEM.

that the Rh1 that accumulates in Rab7 compartment is targeted for degradation, thus leading to retinal degeneration (*Chinchore et al., 2009*). Accumulation of Rh1 in Rab7-positive endosomes may explain the progressive microvillar collapse and reduced Rh1 protein levels in the cell body of *dPLD^3.1^*. Both, the microvillar degeneration and reduced PA levels of *dPLD^3.1^* retinae were rescued by a *dPLD* transgene with intact lipase activity and elevation of PA levels was sufficient to rescue this phenotype. Collectively, our observations strongly suggest that photoreceptors depend on a light-activated *dPLD* to generate PA to maintain apical membrane turnover during illumination. They also suggest that protein-protein interactions of PLD, independent of its catalytic activity, may not be a primary mechanism underlying the function of this enzyme in cells.

In principle, the number of RLVs seen in a photoreceptor following illumination is a balance between ongoing endocytosis and processes that remove endocytosed RLVs either by recycling to the microvillar membrane or targeting to the late endosome-lysosome system for degradation. Using the temperature-sensitive allele of dynamin *shi^ts1^*, we were able to uncouple RLV endocytosis from their removal from the cell body and found that the generation of RLVs during illumination was not dependent on *dPLD* activity (*Figure 5I*) and the number of Rab5-positive RLVs was not increased in *dPLD^3.1^* photoreceptors (*Figure 2F*). Collectively, these observations suggest no primary defect in clathrin dependent endocytosis in *dPLD^3.1^*. However, we found that in *dPLD^3.1^*, the clearance of endocytosed RLVs was dramatically slower than in controls implying that dPLD supports a process that clears RLVs from the cell body. These RLVs were Rab7-positive suggesting that they accumulate in late endosomes. Conversely, in *Rh1>dPLD*, the number of Rab7-positive RLVs was fewer than in wild-type cells. Together, these observations strongly suggest that *dPLD* activity supports a process that clears RLVs post endocytosis.

Following endocytosis, endosomes containing trans-membrane proteins (such as Rh1) can be targeted for lysosomal degradation or be retrieved for recycling to other membranes through retromer-dependent processes. The enhanced RLV numbers as well as retinal degeneration in *dPLD^3.1^* could be rescued by enhancing retromer activity and the ability of *dPLD* overexpression to reduce RLV number during illumination required intact retromer activity. Together, these observations suggest that during illumination *dPLD* stimulates RLV clearance through a retromer-dependent mechanism. We found enhanced number of Rab7-positive RLVs in *dPLD^3.1^* and *Rh1>dPLD* had reduced number of Rab7-positive RLVs. A previous study has reported that retromer activity can clear RLVs from a Rab7-positive compartment (*Wang et al., 2014*). Together, our findings suggest that in the absence of dPLD the sorting of RLVs away from Rab7 endosomes into retromer-dependent recycling is inefficient.

Why might cargo sorting in *dPLD^3.1^* be abnormal ? Sorting reactions in vesicular transport often involve a small GTPase working in conjunction with a lipid-metabolizing enzyme. Altering *garz* function, presumably altering Arf1-GTP levels, has three consequences: (i) in wild-type cells, enhancing *garz* levels results in fewer RLVs and blocks the rise in RLVs seen during light exposure. (ii) enhancing *garz* levels reduces RLV accumulation in *dPLD^3.1^* (iii) enhancing *garz* levels suppresses degeneration in *dPLD^3.1^* while depleting *garz* in wild-type flies results in light-dependent retinal degeneration with a time course similar to that seen in *dPLD^3.1^*. Thus, an Arf1-GTP dependent step is required for both RLV turnover and maintaining apical domain size in photoreceptors. Our finding that the ability of *Rh1>dPLD* to modulate RLV number requires intact *garz* function (*Figure 8A*) is consistent with this model. These findings imply that Arf1-GTP levels positively regulate a step that enhances RLV recycling to the microvillar plasma membrane in the face of ongoing light-induced endocytosis, presumably through retromer complex activity. In support of this idea, we found that the ability of *Rh1>garz* to reduce RLV numbers during illumination depends on intact retromer function

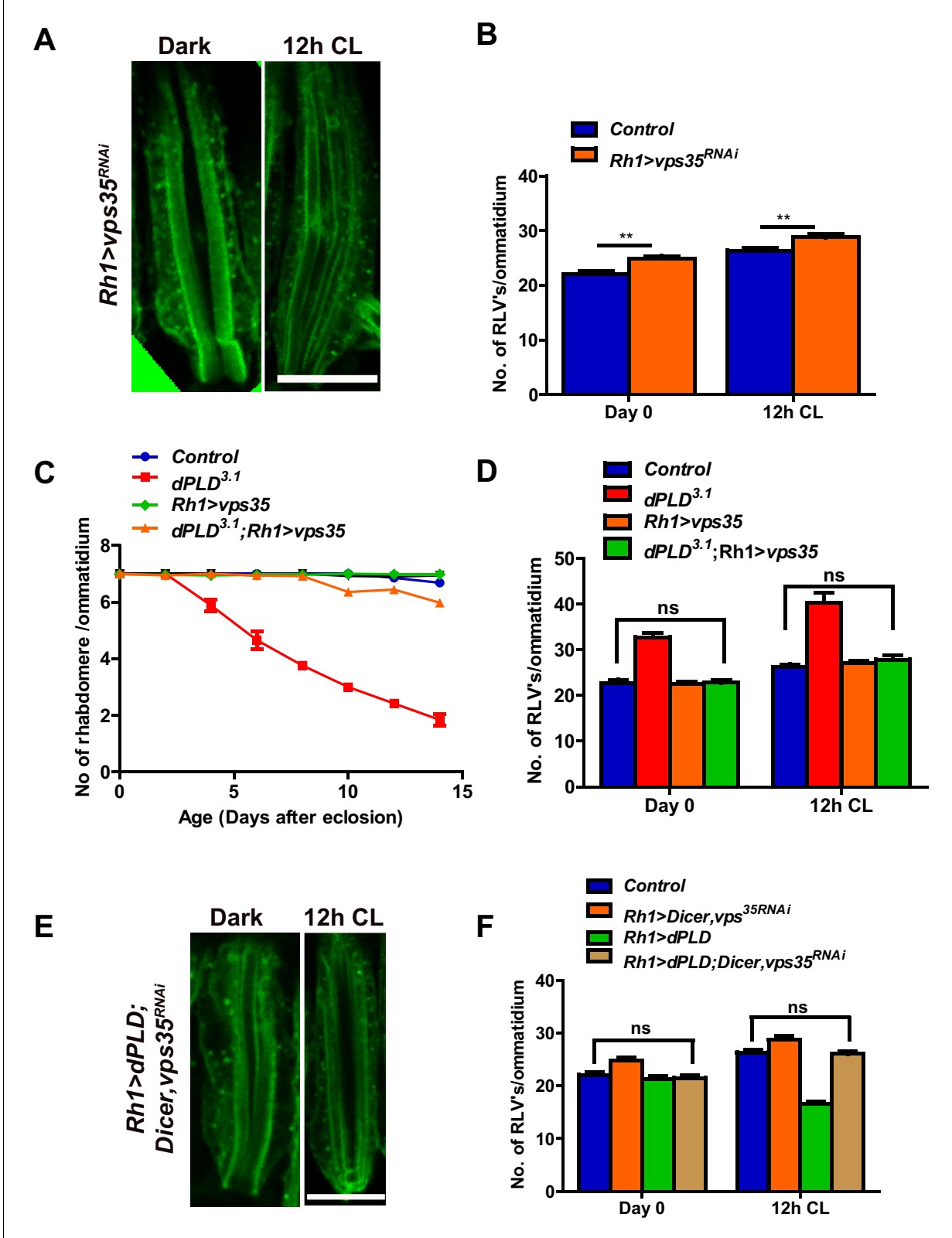

**Figure 6.** dPLD regulates clearance of RLVs via retromer function. (**A**) LS of retinae stained with Rh1 from *Rh1>Dicer,vps35^RNAi*. Rearing conditions are indicated at the top of panels. Scale bar:5 µm. (**B**) Quantification of RLVs from LS of retinae *control* and *Rh1>Dicer,vps35^RNAi*. n = 10 ommatidia taken from three separate preps. (**C**) Quantification of retinal degeneration in *control*, *dPLD^3.1*, *Rh1>vps35* and *dPLD^3.1;Rh1>vps35*. n = 50 ommatidia taken from at least five separate flies. (**D**) Quantification of RLVs from LS of retinae from *control*, *dPLD^3.1*, *Rh1>vps35* and *dPLD^3.1;Rh1>vps35*. n = 10

*Figure 6 continued on next page*

*Figure 6 continued*
ommatidia taken from three separate preps. (E) Longitudinal section of retinae stained with Rh1 *Rh1>dPLD; Dicer,vps35^RNAi*. Rearing condition is indicated at the top of each panel. Scale bar: 5 μm. (F) Quantification of RLVs from longitudinal section of retinae from *control, Rh1>Dicer,vps35^RNAi*, *Rh1>dPLD* and *Rh1>dPLD; Dicer,vps35^RNAi*. n = 10 ommatidia taken from three separate preps. Data are presented as mean ± SEM.

(*Figure 8B*). We propose, that during illumination, rhabdomere size is maintained by the balance between clathrin-dependent endocytosis generating RLVs and an Arf1-GTP dependent sorting event that recycles RLVs to the plasma membrane via retromer activity (*Figure 8D*). *dPLD*, specifically its product PA is likely able to balance these two reactions by coupling light-induced endocytosis to Arf1-dependent sorting of RLVs into the recycling pathway. Previous studies have identified proteins from brain cytosol that bind PA *in vitro* and are known to regulate membrane transport events; prominent among these was Arf1 (*Manifava et al., 2001*) although the *in vivo* significance of this binding is unknown. Our findings that the biological activity of Arf1$^{CA}$ in photoreceptors requires intact *dPLD* activity and that the ability of increased *garz* (Arf1-GEF) levels to clear RLVs requires intact *dPLD* function suggests that Arf1 is a key target of PA generated by dPLD in mediating sorting and recycling of RLVs in photoreceptors. It has been reported that EHD1 an ATPase required to generate tubular recycling endosomes is recruited by MICAL-L1 and the BAR domain protein syndapin2 both of which bind PA (*Giridharan et al., 2013*). It is possible that these proteins are also targets of PA generated by dPLD. In the absence of PA, RLV sorting into the recycling pathway is impaired in *dPLD$^{3.1}$*, some fraction of the endocytosed RLVs accumulates in Rab7 endosomes and is targeted for degradation leading to the reduction in Rh1 levels. These reduced Rh1 levels likely account for the reduced light sensitivity of dPLD mutants reported both in this study as well as in a previous analysis (*LaLonde et al., 2005*).

What is the transduction pathway between photon absorption and dPLD activation? *dPLD$^{3.1}$* photoreceptors show normal electrical responses to light and the microvillar degeneration of *dPLD$^{3.1}$* could not be suppressed by a strong hypmorph of *dGq* (*Scott et al., 1995*) that is required for PLCβ dependent phototransduction. We also found that light-activated elevation of PA levels was dependent on dPLD activity but did not require Gq-PLCβ signalling. Collectively, these findings imply that *dPLD* activity is dispensable for Gq-PLCβ mediated activation of TRP channels and that the light-dependent degeneration of *dPLD$^{3.1}$* is not a consequence of abnormal TRP channel activation.

Our findings suggest that M activates dPLD without the requirement of Gq function, although the molecular mechanism remains to be determined. dPLD has been reported to be localized in the submicrovillar cisternae (*LaLonde et al., 2005*; *Raghu et al., 2009*), a specialization of the smooth endoplasmic reticulum that is positioned ca. 10 nm from the plasma membrane at the base of the microvilli (*Yadav et al., 2016*). At this location, dPLD might bind to the C-terminal tail of M either before or after it is endocytosed into RLV; binding of mammalian PLD1 has been reported to the C-terminal tail of several rhodopsin superfamily GPCRs including the 5-HT2$_a$, muscarinic and opioid receptors [(*Barclay et al., 2011*) and references therein]. It is possible that PA produced by dPLD bound to the C-terminus of Rh1 may then stimulate recycling to the apical membrane. Thus, the control of apical membrane turn over by dPLD during illumination may represent an example by which ligand bound GPCRs signal without a direct involvement for heterotrimeric G-protein activity. More generally in the brain,neurons and glial cells express GPCRs (5-HT2a, mGluR and opioid receptors) of key functional importance. Controlling these GPCR numbers on the plasma membrane during receptor stimulation (of which rhodopsin turnover during illumination is a prototypical example) is of critical importance to brain function and mechanisms that regulate this process will likely be crucial for the understanding and treatment of neuropsychiatric syndromes.

## Materials and methods

### Fly cultures and stocks

Flies were reared on medium containing corn flour, sugar, yeast powder and agar along with antibacterial and antifungal agents. Flies were maintained at 25°C and 50% relative humidity. There was no internal illumination within the incubator, and the flies were subjected to light pulses of short

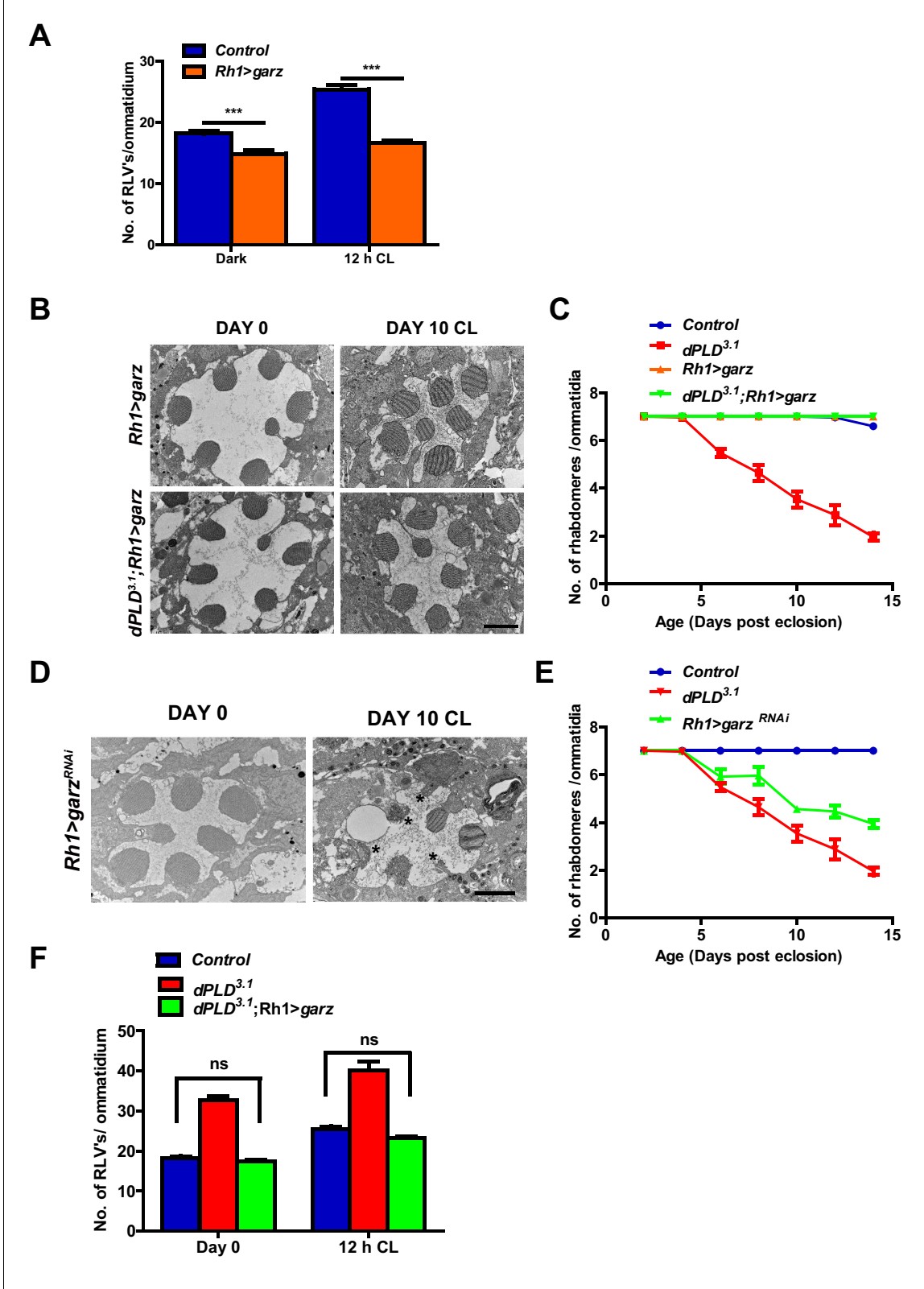

**Figure 7.** Arf1 activity and retinal degeneration in *dPLD^3.* . (**A**) Quantification of RLVs from longitudinal section of retinae from control and *Rh1>garz*. n = 10 ommatidia taken from three separate preps. (**B**) TEM images showing single ommatidium from *Rh1>garz* and *dPLD^3.1^;Rh1>garz* PRs of flies. Rearing condition is indicated on the top of the image. Scale bar: 1 μm. (**C**) Quantification of retinal degeneration in control, *dPLD^3.1^*, *Rh1>garz* and *dPLD^3.1^;Rh1>garz*. n = 50 ommatidia taken from at least five separate flies. (**D**) TEM images showing single ommatidium from *Rh1>garz^RNAi^* PRs of flies.
*Figure 7 continued on next page*

*Figure 7 continued*

Rearing condition is indicated on the image. Scale bar:1 µm. (**E**) Quantification showing the retinal degeneration in control, *dPLD³·¹* and *Rh1>garz^RNAi*. n = 50 ommatidia taken from at least five separate flies. **G**) Quantification of RLVs from longitudinal section of retinae from control, *dPLD³·¹*, *dPLD³·¹*; *Rh1>garz*. n = 10 ommatidia taken from three separate preps. Data are presented as mean ± SEM.

duration only when the incubator door was opened. When required, flies were grown in an incubator with constant illumination from a white light source (intensity ~2000 lux).

The wild type used for all experiments was Red Oregon-R. GAL4-UAS system was used to drive expression of transgenic constructs. The following transgenic lines were obtained from the Bloomington Stock Center: UAS-GFP::Rab5 (B#43336),UAS-YFP::Rab7 (B#23270). UAS-*garz*^RNAi (V# 42140) was obtained from the Vienna Drosophila RNAi Center. Dicer;UAS-vps35^RNAi was obtained from Miklós Sass ( Eötvös Loránd University, Budapest, Hungary) and UAS-vps35::HA was obtained from Prof. Hugo Bellen (Baylor College of Medicine, Howard Hughes Medical Institute, Houston).

## Optical neutralization and scoring retinal degeneration

Flies were cooled on ice, decapitated using a sharp blade, and fixed on a glass slide using a drop of colorless nail varnish. Imaging was done using 40X oil objective of Olympus BX43 microscope. In order to obtain a quantitative index of degeneration, atleast five flies were scored for each time point. A total of 50 ommatidia were assessed to generate degeneration index. To quantify degeneration, a score of one was assigned to each rhabdomere that appeared to be wild type. Thus, wild-type ommatidia will have a score of 7. Mutants undergoing degeneration will have a score between 1 and 7. Score were expressed as mean ± SEM.

## Electroretinograms

Flies were anesthetized on ice and immobilized at the end of a disposable pipette tip using a drop of nail varnish. The recording electrode (GC 100 F-10 borosilicate glass capillaries, 1 mm O.D and 0.58 mm I.D from Harvard apparatus filled with 0.8% w/v NaCl solution) was placed on the surface of eye and the reference electrode was placed on the neck region/thorax. Flies were dark adapted for 5 min followed by ten repeated green light flashes of 2 s duration, each after an interval of 10 s. Stimulating light was delivered from a LED light source placed within a distance of 5 mm of the fly's eye through a fiber optic guide. Calibrated neutral density filters were used to vary the intensity of the light over five log units. Voltage changes were amplified using a DAM50 amplifier (WPI) and recorded using pCLAMP 10.2. Analysis of traces was performed using Clampfit (Axon Laboratories).

## Western blotting

Heads from 1-day-old flies (unless otherwise specified) were decapitated in 2X SDS-PAGE sample buffer followed by boiling at 95°C for 5 min. For detection of rhodopsin, samples were incubated at 37°C for 30 min and then subjected to SDS-PAGE and western blotting. The following antibodies were used: anti-rhodopsin (1:250-4C5), anti-α-tubulin (1:4000,E7c), anti-TRP (1:4000) and anti-NORPA (1:1000). All secondary antibodies (Jackson Immunochemicals) were used at 1:10000 dilution. Quantification of the blot was done using Image J software from NIH (Bethesda, MD, USA).

## Immunohistochemistry

For immunofluorescence studies retinae from flies were dissected under low red light in phosphate buffer saline (PBS). Retinae were fixed in 4% paraformaldehyde in PBS with 1 mg/ml saponin at room temperature for 30 min. Fixed eyes were washed three times in PBST (1X PBS + 0.3% TritonX-100) for 10 min. The sample was then blocked in a blocking solution (5% Fetal Bovine Serum in PBST) for 2 hr at room temperature, after which the sample was incubated with primary antibody in blocking solution overnight at 4°C on a shaker. The following antibodies were used: anti-Rh1 (1:50), anti-TRP (1:250) and anti-GFP (1:5000,abcam [ab13970]). Appropriate secondary antibodies conjugated with a fluorophore were used at 1:300 dilutions [Alexa Fluor 488/568/633 IgG, (Molecular Probes)] and incubated for 4 hr at room temperature. Wherever required, during the incubation with secondary antibody, Alexa Fluor 568-phalloidin (Invitrogen) was also added to the tissues to stain

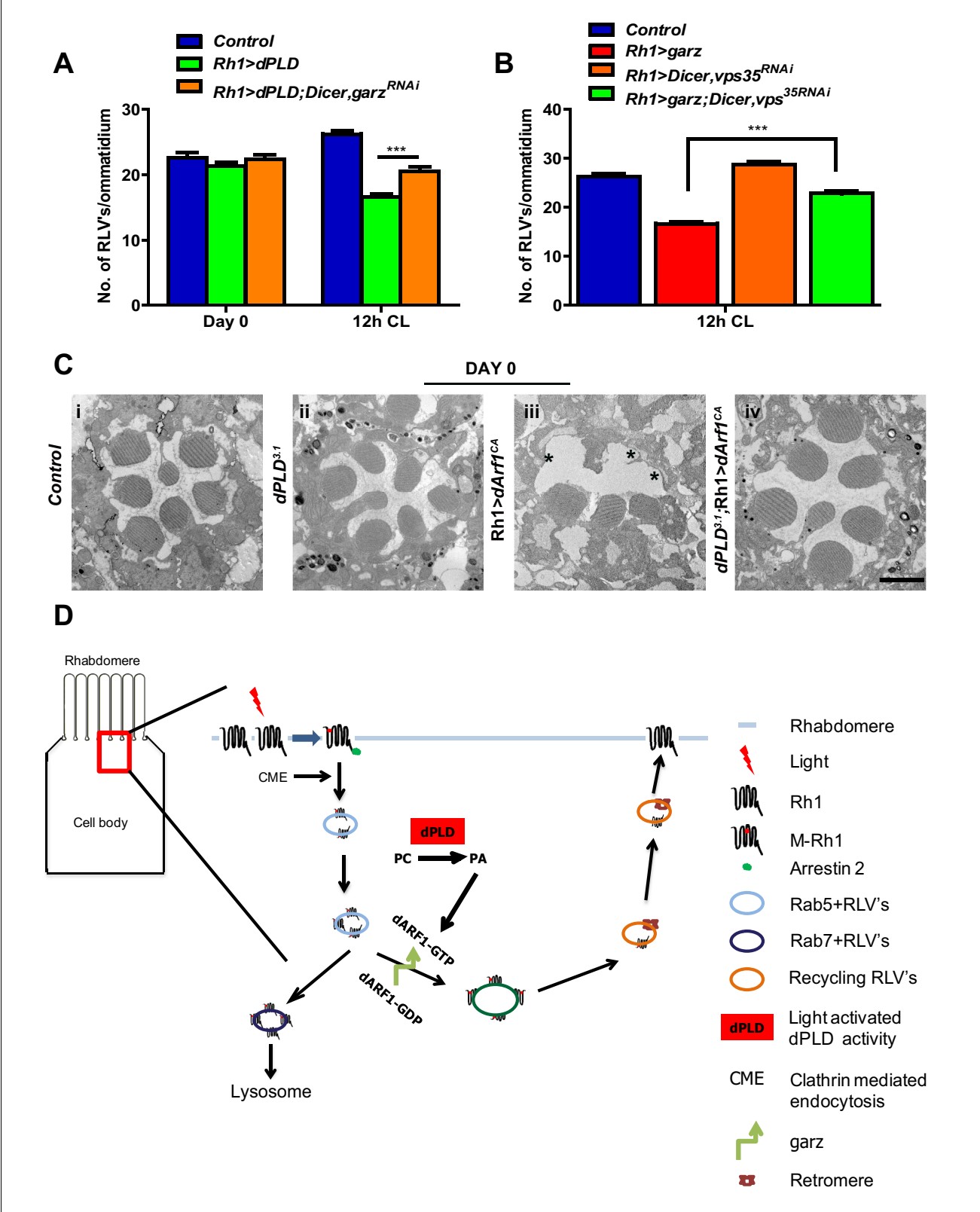

**Figure 8.** *dPLD* and *garz* are required for RLV clearance during illumination. (**A**) Quantification of RLVs from longitudinal section of retinae from control, *Rh1>dPLD*, *Rh1>dPLD; Dicer,Rh1>garz^RNAi^*. n = 10 ommatidia taken from three separate preps. (**B**) Quantification of RLVs from longitudinal section of retinae from control, *Rh1>garz*, *Rh1>Dicer,vps35^RNAi^* and *Rh1>garz;Dicer,vps35^RNAi^*. n = 10 ommatidia taken from three separate preps. (**C**) TEM images showing single ommatidium from control and *dPLD^3.1^*,Rh1>*Arf1^CA^* and *dPLD^3.1^*;Rh1>*Arf1^CA^* PRs of day 0-old flies post-eclosion. Scale bar: *Figure 8 continued on next page*

*Figure 8 continued*

1 µm (**D**) A model of the light activated turnover of rhabdomere membranes in *Drosophila* photoreceptors. The cross-section of a PR is shown. The area indicated by the red box is enlarged to the left. PC-phosphatidylcholine, PA-phosphatidic acid, dARF1-GTP- GTP bound active ARF1, dARF1-GDP-GDP bound inactive Arf1, brown star indicates retromer, blue RLVs indicate endocytic compartment while orange RLVs indicate recycling compartment.

the F-actin. After three washes in PBST, sample was mounted in 70% glycerol in 1X PBS. Whole mounted preps were imaged on Olympus FV1000 confocal microscope using Plan-Apochromat 60x, NA 1.4 objective (Olympus).

## Rhodopsin-loaded vesicles (RLV's) counting

Whole mount preparations of photoreceptors stained with anti-Rh1 were imaged on Olympus FV1000 confocal microscope using Plan-Apochromat 60X, NA 1.4 objective (Olympus). The RLV's per ommatidium were counted manually across the Z-stacks using Image J software from NIH (Bethesda, MD, USA).

## Electron microscopy and volume fraction analysis

Samples for TEM were prepared as mentioned in previous publication (*Garcia-Murillas et al., 2006*). Briefly samples were bisected in ice cold fixative solution (For 1 ml: 0.5 ml of 0.2 M PIPES (pH:7.4),80 µl of 25% EM grade glutaraldehyde, 10 µl of 30% $H_2O_2$ and 0.41 ml water). After overnight fixation at 4°C, samples were washed in 0.1M PIPES (thrice 10 min. each) and then fixed in 1% osmium tetroxide (15 mg Potassium Ferrocyanide, 500 µl 0.2M PIPES, 250 µl 4% Osmium tetroxide and 250 µl of distilled water) for 30 min. The eyes were then washed with 0.1M PIPES (thrice 10 min. each) and then stained in en-block (2% Uranyl acetate) for 1 hr. Eyes were dehydrated in ethanol series and embedded in epoxy. Ultrathin sections (60 nm) were cut and imaged on a Tecnai G2 Spirit Bio-TWIN (FEI) electron microscope.

For volume fraction analysis, TEM images of *Drosophila* retinae were acquired and analyzed using the ADCIS Stereology toolkit 4.2.0 from the Aperio Imagescope suite. A grid probe was used whose probe intersections were accurate to about 200–300 points. The volume fraction ($V_f$) of the rhabdomere with respect to its corresponding photoreceptor cell was calculated as:

$$V_f = \frac{\text{Number of points falling on the rhabdomere}}{\text{Total number of points on the cell}}$$

Volume fractions were calculated separately for R1–R6 and R7.

## Scoring retinal degeneration using TEM

TEM images were acquired using Tecnai G2 Spirit Bio-TWIN (FEI) electron microscope. To quantify degeneration, a score of one was assigned to each rhabdomere that appeared to be wild type and a score of 0.5 was assigned to each rhabdomere that appeared to be partially degenerated.

## Isolation of pure retinal tissue

Pure preparations of retinal tissue were collected using previously described methods (*Fujita et al., 1987*). Briefly, 0- to 12-hr-old flies (unless otherwise specified) were snap frozen in liquid nitrogen and dehydrated in acetone at −80°C for 48 hr. The acetone was then drained off and the retinae dried at room temperature. They were cleanly separated from the head at the level of the basement membrane using a scalpel blade.

## Lipid extraction and mass spectrometry

Ten heads or 100 retinae per sample (dissected from 1-day-old flies) were homogenized in 0.1 ml methanol containing internal standards) using an automated homogenizer. The methanolic homogenate was transferred into a screw-capped tube. Further methanol (0.3 ml) was used to wash the homogenizer and was combined in the special tube. 0.8 ml chloroform was added and left to stand for 15 min. 0.88% KCl (0.4 ml) was added to split the phases. The lower organic phase containing

the lipids were dried, re-suspended in 400 µl of chloroform:methanol 1:2 and was ready for analysis. Total lipid phosphate was quantified from each extract prior to infusion into the mass spectrometer.

Mass spectrometer analyses were performed on a LTQ Orbitrap XL instrument (Thermo Fisher Scientific) using direct infusion method. Stable ESI-based ionization of glycerophospholipids was achieved using a robotic nanoflow ion source TriVersa NanoMate (Advion BioSciences) using chips with the diameter of spraying nozzles of 4.1 µm. The ion source was controlled by Chipsoft 8.3.1 software. Ionization voltages were +1.2 kV and −1.2 kV in positive and negative modes, respectively; back pressure was set at 0.95 psi in both modes. The temperature of ion transfer capillary was 180°C. Acquisitions were performed at the mass resolution $R_{m/z400}$ = 100,000. Dried total lipid extracts were re-dissolved in 400 µl of chloroform:methanol 1:2. For the analysis, 60 µl of samples were loaded onto 96-well plate (Eppendorf) of the TriVersa NanoMate ion source and sealed with aluminum foil. Each sample was analyzed for 20 min in positive ion mode where PC was detected and quantified. This was followed by an independent acquisition in negative ion mode for 20 min where PA was detected and quantified.

Lipids were identified by LipidXplorer software by matching *m/z* of their monoisotopic peaks to the corresponding elemental composition constraints. Molecular Fragmentation Query Language (MFQL) queries compiled for all the aforementioned lipid classes. Mass tolerance was 5 p.p.m. and intensity threshold was set according to the noise level reported by Xcalibur software (Thermo Scientific).

## Transphosphatidylation assay

One-day-old flies were starved for 12 hr and then fed on 10% ethanol in sucrose for 6 hr. Following this lipids were extracted (with appropriate internal standards) and phosphatidylethanols detected and quantified by HPLC/MS method (*Wakelam et al., 2007*).

## Data analysis

Data were tested for statistics using unpaired t-test. *** denotes $p<0.001$; ** denotes $p<0.01$; * denotes $p<0.05$ and ns denotes not significant.

# Acknowledgements

This work was supported by the National Centre for Biological Sciences-TIFR, the Biotechnology and Biological Sciences Research Council, UK, Department of Biotechnology, Government of India and a Wellcome Trust-DBT India Alliance Senior Fellowship to PR. We thank the NCBS- Max Planck Lipid center, Electron Microscopy, Imaging (CIFF) and Fly Facility for assistance. We thank Hanneke Okkenhaug for assistance at an early stage of this project. We acknowledge the generous support of numerous colleagues who provided us with fly stocks and antibodies.

# Additional information

### Funding

| Funder | Grant reference number | Author |
| --- | --- | --- |
| Wellcome Trust DBT India Alliance | IA/S/14/2/501540 | Rajan Thakur<br>Nikita Raj<br>Shweta Yadav<br>Sruthi Balakrishnan<br>Bishal Basak<br>Raghu Padinjat |
| National Centre for Biological Sciences | core | Rajan Thakur<br>Aniruddha Panda<br>Nikita Raj<br>Shweta Yadav<br>Sruthi Balakrishnan<br>Bishal Basak<br>Renu Pasricha<br>Raghu Padinjat |
| Department of Biotechnology, | BT/PR4833/MED/30/744/ | Rajan Thakur |

| Ministry of Science and Technology | 2012 | Aniruddha Panda<br>Nikita Raj<br>Shweta Yadav<br>Sruthi Balakrishnan<br>Bishal Basak<br>Renu Pasricha<br>Raghu Padinjat |
| --- | --- | --- |
| Biotechnology and Biological Sciences Research Council | core | Elise Coessens<br>Qifeng Zhang<br>Plamen Georgiev<br>Michael JO Wakelam<br>Nicholas T Ktistakis |

The funders had no role in study design, data collection and interpretation, or the decision to submit the work for publication.

## Author contributions

RT, Conception and design, Acquisition of data, Analysis and interpretation of data, Drafting or revising the article; AP, SY, PG, Acquisition of data, Analysis and interpretation of data, Drafting or revising the article; EC, Generated the dPLD knockout by homologous recombination and performed molecular characterization of the dPLD knockout, Acquired data on the molecular features of the knockout; NR, SB, QZ, Acquisition of data, Analysis and interpretation of data; BB, Contributed unpublished essential data or reagents; RP, Supervised the collection and analysis of the volume fraction data; MJOW, Conception and design for the PtDEOH measurements in addition to analysis and interpretation; NTK, Conception and design, Drafting or revising the article; PR, Conception and design, Analysis and interpretation of data, Drafting or revising the article

## Author ORCIDs

Rajan Thakur, http://orcid.org/0000-0002-9511-5729
Nicholas T Ktistakis, http://orcid.org/0000-0001-9397-2914
Padinjat Raghu, http://orcid.org/0000-0003-3578-6413

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
