## [Decision Letter]

Thank you for submitting your article "Phospholipase D activity couples plasma membrane endocytosis with retromer dependent recycling" for consideration by *eLife*. Your article has been reviewed by two peer reviewers, and the evaluation has been overseen by Randy Schekman as the Reviewing and Senior Editor. The following individual involved in review of your submission has agreed to reveal his identity: Vytas Bankaitis (Reviewer #2).

The reviewers have discussed the reviews with one another and the Reviewing Editor has drafted this decision to help you prepare a revised submission.

Summary:

In this paper Thakur et al. describe a role for phospholipase D (PLD) in regulating membrane turnover in *Drosophila* photoreceptor neurons. The paper makes a solid argument that re-cycling of rhodopsin relies upon a retromer-dependent, PLD-dependent mechanism. Here phosphatidic acid generated by light-activated PLD (via a pathway independent of trimeric G protein activation) is proposed to activate Arf1 and this in turn regulates recycling of the rhodopsin back to the rhabdomere membrane. The significance of this work is that it provides the first confident description of a PLD phenotype in a multicellular eukaryote and in a quantifiable and powerful system of intense lipid signaling. The work is generally thorough, well-controlled and brings new perspective to a field of lipid signaling that badly needs it.

Essential revisions:

1) The genetics would clearly be strengthened by an additional allele or two of PLD and, most importantly, the authors must show that PLD is a null, a significant concern given the molecular nature of the mutation. Minimally this can be done by comparing the effects of homozygotes to one in which *dPLD^3.1^* mutation is placed in trans to a deletion removing the gene.

2) The approaches taken are powerful and the conclusions are generally well-supported. What is missing, however, is a demonstration of what component is PA-activated. This is an important issue as roles for PLD in regulating endosomal trafficking and membrane biogenesis have been demonstrated in yeast. Although the authors devote considerable discussion to the issue, they should address it experimentally. garz seems a likely candidate. It should not be necessary to conduct a full characterization of target but it would be worthwhile for the authors to at least assess whether garz, any fly retromer components, or fly ARF itself bind PA. there are already clues that mammalian ARF binds PA so a likely candidate is already known.

3) The genetic epistasis data argue PLD activates Arf and not the other way around (as dogma has long held). This point should be emphasized and it reinforces the importance of defining the target of regulation by PA as outlined in (a).

4) Have the authors conducted a stimulation train where they compare response run-down in wt vs. pld retina? They report a single burst ERG, but the run-down would be a more informative experiment.

5) The fact that several independent means for elevating PA result in suppression of pld defects argues that PLD protein::protein interactions are not functionally important. The authors should emphasize this point more forcefully as there is a large literature on the PLD interactome with much speculation on what it might mean.

6. In the subsection “dPLD activity supports RLV removal from the cell body during illumination”, last paragraph, first sentence – please provide the reference.

[Editors' note: further revisions were requested prior to acceptance, as described below. There is no accompanying Author response.]

Thank you for resubmitting your work entitled "Phospholipase D activity couples plasma membrane endocytosis with retromer dependent recycling" for further consideration at *eLife*. Your revised article has been favorably evaluated by Randy Schekman as the Senior and Reviewing Editor, and two reviewers.

Your manuscript has been improved but the reviewers feel that a bit more work is needed to justify your conclusions. The Spo20 control experiment shows that the liposome binding of this protein is specific to PA (as is expected) and that it is concentration-dependent. However, you should examine dARF binding over the same range of concentration of PtdOH rather than showing one concentration in one Figure panel (8D) and another in the supplemental data. In addition, you should include a PtdSer control as well. Although one reviewer felt that the PtdOH binding data being omitted, the consensus was that you should perform the additional experiment as suggested above. This should be straightforward and it would lend biochemical support to your strong genetic/biological data.

Full comments of the reviewers:

*Reviewer #1:*

The response to my concerns about the genetics and additional alleles is satisfactory. I am skeptical about the biochemistry showing binding of Arf1 to PA. Several issues concern me: 1) The binding is weak, as they indicate; 2) the experiments are not well described; and 3) the discrepancy between the behavior of Spo2 (positive control) binding at different ratios of POPC/PA and Arf1 vs POPC/PA are problematic. The binding of at 70:30 is optimal for Spo2 and at 90:10 the binding is weak. The data in Figure 8 shows binding at 90:10. What was the binding at different concentrations? I will leave it to the lipid biochemists to give it their expert evaluation but I am skeptical of the data. It looks too different from the positive control to be compelling. I would prefer the authors leave the issue of the protein binding to PA unresolved than presenting data of this quality.

*Reviewer #2:*

The authors have done a reasonable job in positively addressing the comments raised in the initial review. They have revised the text of the manuscript to better highlight what this reviewer considers to be outstanding points made by the data. While it would have been ideal to pin-point a clear target for PLD-generated PtdOH in regulating Arf or the retromer complex, the likelihood of multiple targets does make this potentially an unattainable result in the time frame allotted. The fact that Arf does bind PtdOH provides biochemistry that is at least consistent with the pathway laid out by the genetics. In summary, this is an important paper and I am satisfied with the revisions made.

---

## [Author Response]

[…]

*Essential revisions:*

*1) The genetics would clearly be strengthened by an additional allele or two of PLD and, most importantly, the authors must show that PLD is a null, a significant concern given the molecular nature of the mutation. Minimally this can be done by comparing the effects of homozygotes to one in which dPLD^3.1^ mutation is placed in trans to a deletion removing the gene.*

We studied the effect of dPLD^3.3^, an independently isolated allele of dPLD that was also generated by homologous recombination; its molecular nature at the dPLD locus is identical to that of *dPLD^3.1^* A trans-heterozygote combination of *dPLD^3.1^*/dPLD^3.3^ was compared to *dPLD^3.1^*/*dPLD^3.1^*. The light-dependent retinal degeneration phenotype of these two genotypes was compared. The severity and time course of retinal degeneration was found to be similar in both genotypes. This data is presented in Figure 3—figure supplement 1.

We also studied the phenotype of *dPLD^3.1^*/*dPLD^3.1^* in comparison with *dPLD^3.1^/*Df (2R)ED1612. Df (2R) ED1612 is a Drosdel deletion of the region of the *dPLD* gene and removes the full sequence of *dPLD* (Figure 2—figure supplement 1). *dPLD^3.1^*/Df (2R)ED1612 also shows light-dependent retinal degeneration over a similar time course to *dPLD^3.1^/dPLD^3.1^*. Importantly, the severity of degeneration in *dPLD^3.1^*/Df (2R)ED1612 was *not* greater than in *dPLD^3.1^*/*dPLD^3.1^* (Figure 3—figure supplement 1) providing additional support that *dPLD^3.1^* is a null allele. The slightly slower rate of degeneration in *dPLD^3.1^*/Df (2R)ED1612 is likely due to the reason that Df (2R)ED1612, removes, in addition to dPLD, 115 other genes one or more of which may have a modifier effect on the dPLD phenotype.

*2) The approaches taken are powerful and the conclusions are generally well-supported. What is missing, however, is a demonstration of what component is PA-activated. This is an important issue as roles for PLD in regulating endosomal trafficking and membrane biogenesis have been demonstrated in yeast. Although the authors devote considerable discussion to the issue, they should address it experimentally. garz seems a likely candidate. It should not be necessary to conduct a full characterization of target but it would be worthwhile for the authors to at least assess whether garz, any fly retromer components, or fly ARF itself bind PA. there are already clues that mammalian ARF binds PA so a likely candidate is already known.*

We agree that the molecular mechanism by which PA exerts its effects is an important issue. However, it is also a challenging one given that there are dozens of proteins that have been shown to bind PA in vitro. In the context of the present study, Arf1, garz and retromer components are plausible protein targets to which PA might bind and exert its effects. Given the large number of proteins in the retromer complex it was difficult to decide on which ones to study. However we performed in vitro liposome based binding assays to test PA binding to endogenous *Drosophila* Arf1 and GARZ. We could not demonstrate binding of garz to PA containing liposomes (Figure 8—figure supplement 1C). However, we were able to demonstrate weak binding of endogenous *Drosophila* Arf1 to PA containing liposomes (Figure 8). The ability of Arf1 to bind to PA containing liposomes was weak in comparison to that of the well-known PA binding protein Spo20. Up to 30% PA was required to show good binding of Arf1 to PA containing liposomes. Together with our finding that the biological effects of expressing Arf1CA on photoreceptors required intact dPLD function (Figure 8), the in vitro binding assay suggests that Arf1 is a likely protein target of PA produced by PLD.

*3) The genetic epistasis data argue PLD activates Arf and not the other way around (as dogma has long held). This point should be emphasized and it reinforces the importance of defining the target of regulation by PA as outlined in (a).*

This has been reinforced in the Discussion.

*4) Have the authors conducted a stimulation train where they compare response run-down in wt vs. pld retina? They report a single burst ERG, but the run-down would be a more informative experiment.*

We have now included two additional physiological protocols (i) Photoreceptors were exposed to a continuous 10 s flash of light to check for any run down/loss of response. This was not seen; the raw traces and quantified data is now included as Figure 2—figure supplement 3. (ii) We also stimulated photoreceptors with a train of rapid flashes at the high light intensity. Once again there was no difference between the response of *dPLD^3.1^* and controls; both continued to respond in an equivalent way with no loss of electrical response. Representative traces and quantification are now included as Figure 2—figure supplement 3.

*5) The fact that several independent means for elevating PA result in suppression of pld defects argues that PLD protein::protein interactions are not functionally important. The authors should emphasize this point more forcefully as there is a large literature on the PLD interactome with much speculation on what it might mean.*

We have attended to this in the text.

*6. In the subsection “dPLD activity supports RLV removal from the cell body during illumination”, last paragraph, first sentence – please provide the reference.*

The relevant reference Chinchore et al., 2009 has been added.